Status and trends of giant clam populations demonstrate the effectiveness of village-based protection in American Sāmoa

Marra-Biggs Paolo Paolomb@Hawaii.edu 1
Brown Eric K. 2
Ochavillo Domingo Galgo 3
Green Alison L. 4
Lawrence Alice 5
Tramonte Carlos 1
Vaeoso Valentine 2
Moffitt Ian 6
Schnurle Kersten 7
Molina Nury 8
Toonen Robert J. 1
1 Hawaiʻi Institute of Marine Biology, School of Ocean & Earth Sciences & Technology, University of Hawaii at Mānoa , Kāne‘ohe , Hawaiʻi , United States of America
2 National Park of American Samoa, US National Park Service , Pago Pago , American Samoa , American Samoa
3 Department of Marine and Wildlife Resources, American Samoan Government , Pago Pago , American Samoa , American Samoa
4 Alison Green Marine , Gold Coast , Queensland , Australia
5 Bangor University , Bangor , Wales , United Kingdom
6 Long Marine Lab, University of California, Santa Cruz , Santa Cruz , CA , United States of America
7 Oregon Institute of Marine Biology, University of Oregon , Charleston , OR , United States of America
8 Marine Science Institute, University of California, Santa Barbara , Santa Barbara , CA , United States of America
Yapıcı Sercan
Electronic publication date: 2025 Nov 14
Publication date: 2025
Volume: 13
Electronic Location ID: e20290
Received 2025 May 19; Accepted 2025 Oct 3
Copyright: ©2025 Marra-Biggs et al.
Copyright year: 2025
Copyright holder: Marra-Biggs et al.
License: This is an open access article distributed under the terms of the Creative Commons Attribution License, which permits unrestricted use, distribution, reproduction and adaptation in any medium and for any purpose provided that it is properly attributed. For attribution, the original author(s), title, publication source (PeerJ) and either DOI or URL of the article must be cited.
License URL: https://creativecommons.org/licenses/by/4.0/

Keywords: Tridacna, Population assessment, ESA, Community management areas, Village protected areas, Traditional stewardship, Marine protected areas, Endangered species assessment, Community-based management, Tridacna population dynamics

Funding: Hawaiʻi Sea Grant, NOAA: Hawaiʻi Sea Grant #NA22OAR4170108, PRoject No. R/HE-44 Pacific Islands Climate Adaptations Science Center Colonel Willys E. Lord and Sandina L. Lord Scholarship, Robinson Fellowship, University of Hawaiʻi, Ruth D. Gates Scholarship, Watson T. Yoshimoto Fellowship National Park of American Samoa P22AC02332 The American Samoan Government’s Department of Marine and Wildlife Resources NPSA This project was possible via funding sources from Hawaiʻi Sea Grant, NOAA: Hawaiʻi Sea Grant (#NA22OAR4170108, Project No. R/HE-44), Pacific Islands Climate Adaptations Science Center, Colonel Willys E. Lord and Sandina L. Lord Scholarship, Robinson Fellowship, University of Hawaiʻi, Ruth D. Gates Scholarship, Watson T. Yoshimoto Fellowship, National Park of American Samoa (P22AC02332), and the American Samoan Government’s Department of Marine and Wildlife Resources. Personnel from NPSA helped edit and revise the manuscript. The funders had no role in study design, data collection and analysis, decision to publish, or preparation of the manuscript.

==============================
Giant clams (subfamily Tridacninae) serve diverse ecological functions in coral reef ecosystems but have experienced severe populatiaon declines across much of their native ranges. Continued overharvesting, habitat degradation, and climate change impacts reinforce the need for updated population assessments and have prompted consideration for endangered species status. Here, we report a territory-wide evaluation of giant clam populations in American Sāmoa, integrating historical data (1994/95, 2002, and 2018) with new surveys conducted from 2022 to 2024 to assess the population status of these ecologically important bivalves. Using belt transects (50 m × 2 m at 10 m depth), we examined clam densities, size-class distributions, species composition, and population trends across six islands—Tutuila, Aunuʻu, Ofu, Olosega, Taʻū, and Muliāva. This study added 264 transects to the historical dataset and showed population densities of giant clams varied among islands but have remained relatively consistent over time. Using univariate and factorial ANOVAs of giant clam abundance and size data, we assessed spatial and temporal variation across American Sāmoa, and our analyses tested for effects of island, year, protection status, and their interactions. The lowest recent clam densities (83.5 individuals per hectare in 2022) were observed on the main island of Tutuila, where 98% of residents live. Mean clam density on Tutuila has increased from 14.1/ha in 1994/95, but the island has considerable variation amongst locations. Remote islands, such as Taʻū and Muliāva, showed higher densities, up to 812 and 1,166/ha, respectively. Most (96.7%) of giant clams found on transects were identified as Tridacna maxima, with infrequent occurrences of the cryptic species T. squamosa and T. noae, primarily within specific, village-managed protected areas. On Tutuila, surveys sites included a variety of jurisdictions and levels of management, with village protected areas and remote sites supporting both higher clam densities and larger individuals. Overall, inaccessible remote sites and those under traditional village enforcement significantly outperformed all other management strategies, including federally designated no-take zones. These findings suggest that empowering traditional Indigenous community stewardship may offer a viable alternative to blanket federal restrictions, and support the importance of localized, community-based management practices in American Sāmoa. We emphasize the need for more frequent monitoring across varying depths, anthropogenic influences, and management regimes to better understand the population dynamics of these valuable coral reef species.

Introduction

Giant clams provide important ecological benefits in coral reef systems, contributing to reef productivity and benefitting species at all trophic levels (Neo et al., 2015a). Tridacna species derive nutrition either through filter feeding (Guest et al., 2008), photosynthetic algae living symbiotically in their tissues (Munro, 2005), or both (Fatherree, 2006; Toonen et al., 2012). This mixotrophic flexibility allows them to thrive in both clear, oligotrophic waters and turbid, nutrient-rich lagoonal areas. Additionally, clams filter water of organic detritus, inorganic nitrogen, and pollutants (Fitt, Heslinga & Watson, 1993), increasing visibility, reducing nutrient loads, and promoting the growth of other photosynthetic organisms such as corals (Edinger et al., 2000; Neo et al., 2015b). Giant clams also potentially prevent eutrophic algal blooms by fixing organic particulate matter following large terrestrial inputs (Officer, Smayda & Mann, 1982; Heslinga & Fitt, 1987; Hily, 1991; Waters, Story & Costello, 2013; Neo et al., 2015a; Zhang et al., 2023). As ecosystem engineering species, they provide settlement surfaces for epibionts such as macroalgae, sponges, and corals, forming communities dependent on their provided habitat, making them ecologically important to a wide range of reef organisms (Fatherree, 2006; Mekawy, 2014; Neo et al., 2015a). Yet, despite their crucial ecological role, giant clams face escalating threats that have led to widespread population declines and heightened conservation concerns (Meadows, 2016; Lee et al., 2022; Dolorosa et al., 2024).

Recognizing their vulnerable status, all species of giant clams are of conservation concern and currently listed under the Convention on International Trade in Endangered Species of Wild Fauna and Flora (CITES) and the International Union for Conservation of Nature (IUCN) (Tisdell, 1989; Wells, 1997; Neo et al., 2017). Overharvesting is widely attributed as the primary cause for the low abundances of these species, and since the IUCN evaluation in 1996, populations have continued to decline across their range (Meadows, 2016; Lee et al., 2022; Dolorosa et al., 2024). In 2024, the IUCN reassessed all giant clam species, updating their Red List classifications to reflect ongoing threats such as overexploitation and habitat degradation (Rippe et al., 2024). Tridacna gigas now ranks as critically endangered, while T. derasa and T. mbalavuana are listed as endangered, and Hippopus hippopus and H. porcellanus are considered vulnerable. Most remaining species (T. crocea, T. noae, T. maxima, T. squamosina) are categorized as least concern, though T. rosewateri and the recently described T. elongatissima remain data deficient due to limited information.

In response to these continued declines, a petition was filed in 2016 to list all species of Tridacninae giant clams (excluding Tridacna rosewateri, due to a lack of information) as threatened or endangered under the Endangered Species Act (ESA) pursuant to section ‘Discussion’ (b) of the ESA. In July 2024, the NOAA National Marine Fisheries Service completed a status review for seven species (Hippopus hippopus, H. porcellanus, Tridacna derasa, T. gigas, T. mbalavuana, T. squamosa, and T. squamosina) and “determined that the most significant threats to H. hippopus, T. derasa, and T. gigas are overexploitation and inadequacy of regulatory mechanisms to address overutilization” but that “T. squamosa is at low risk of extinction throughout all of its range” (Rippe et al., 2024). If the proposed listings are finalized, six of the seven species would be classified as “threatened” under the ESA, prompting prohibitions on import, export, and interstate commerce without a permit (Rippe et al., 2024). Additionally, four species (T. crocea, T. maxima, T. noae, and T. squamosa) are proposed for listing as threatened due to their “similarity of appearance” with each other, which would result in the prohibition of the import and export of derivative products (raw tissue, shells, carvings, jewelry, pearls) from these species.

Tridacna species are particularly vulnerable to overharvesting due to their sensitive life history traits. These clams are slow-growing, have relatively late reproductive maturity, and are sedentary as adults, making them easy targets for exploitation and local extirpation (Craig, 2009; Neo et al., 2015b; Meadows, 2016). Evaluating the population stability of giant clam species requires measuring the effective population size that is determined by spatial aggregations and abundance across size classes at each location (Levitan, 2005; Guy, Smyth & Roberts, 2019). For example, other mollusk species such as oysters showed decreased brooding rates as distances between conspecifics increased over 1.5 m apart (Guy, Smyth & Roberts, 2019), and complete fertilization failure was found in mussels when densities were below 10 mussels/m2 (Downing et al., 1993). For a multitude of reasons, giant clam populations are susceptible to the Allee effect, an ecological phenomenon where individual fitness and population growth decline at low population densities, making it harder for the population to recover (Stephens & Sutherland, 1999; Donahue, 2006). Fertilization success in adults relies on the reception of chemical cues to activate spawning synchrony, and larval dispersal is driven primarily by ocean currents (Neo et al., 2013). Larvae and juveniles exhibit high sensory abilities and have strong preference for close proximity, actively moving towards directions of higher conspecific effluents for gregarious settlement (Dumas et al., 2014).

As protandrous hermaphrodites, individuals first mature as males after about 3 years (Ellis, 1997) and become simultaneous hermaphrodites after 9–10 years (Munro, 1993; Chambers, 2008). Because of their slow growth rate of 2–4 cm/year (Beckvar, 1981; Hart, Bell & Foyle, 1998; Chambers, 2008; Toonen et al., 2012), individuals may never successfully produce eggs before reaching the minimum legal size for harvest. This size-selective harvest of larger individuals further exacerbates reproductive issues by altering the natural sex ratio by removing the larger females, thereby reducing successful fertilization. Reproductive success in giant clams is heavily influenced by their size, as there is a direct relationship between clam size and fecundity, indicating that smaller species or individuals require more animals to produce the same number of eggs as larger ones (Ellis, 1997). Braley (1987) reported that 70% of Tridacna gigas individuals that released gametes naturally had neighboring conspecifics within a 9-m radius. Even where some large individuals remain, low population densities isolate them from potential mates, leading to reproductive failure and, in some extreme cases, functional extinction of local populations (Waters, Story & Costello, 2013).

Cryptic species have long confounded population estimates of giant clams (e.g., Penny & Willan, 2014; Su et al., 2014). For example, T. noae can only be identified definitively through soft tissue or genetic characters. In some locations, such as Ningaloo Reef in Australia, previously misidentified populations of T. maxima have been described as the new species T. ningaloo (Penny & Willan, 2014) only to later be genetically confirmed as T. noae (Johnson et al., 2016). Furthermore, broad-scale genetic surveys suggest the existence of additional cryptic species, which have not yet been identified or named (Huelsken et al., 2013; Liu et al., 2020), and identification of live animals in the field based on morphological characters of the shell that is often embedded within the reef matrix is fraught with challenges (Fatherree, 2023). As a result, population estimates for many giant clam species are likely confounded by species misidentification and resulted in the NOAA Fisheries proposal to consider Tridacna squamosa, T. noae, T. maxima, and T. crocea as threatened throughout their range based on the similarity of appearance (Rippe et al., 2024).

Faisua (giant clams in Samoan) are deeply woven into the traditional food systems of American Sāmoa and other Pacific Island cultures. In Sāmoa and other Pacific Island nations, giant clams are valued for their status as food items and their potential in small-scale mariculture, providing both nutrition and economic opportunity (Munro, 1993; Bell et al., 2005). These dual roles, as both a food source and cultural symbol, have made them a focal species in conservation strategies across the Pacific. Similarly, in American Sāmoa, they are considered a delicacy and a valued component of subsistence harvests, often gathered for daily consumption and special occasions (Levine & Allen, 2009; Rippe et al., 2024). Their use in fa‘alavelave (ceremonial obligations), communal feasts, and gift exchanges emphasizes their cultural significance (Levine & Allen, 2009). Yet, despite continued conservation concerns, the population status of all giant clam species across American Sāmoa has not been assessed by any published studies since the surveys conducted by Green & Craig (1999), which was conducted prior to recognition of T. noae (Su et al., 2014), so the authors assumed all visually similar samples were T. maxima (A Green, Pers. comm., January 2025). Since then, genetic research has confirmed the presence of previously unreported T. noae in American Sāmoa (Marra-Biggs et al., 2022). The distribution and abundance of T. noae across American Sāmoa remains unknown, because historical population surveys may have misidentified this species as T. maxima, further confounding population estimates for both species.

American Sāmoa is a territory of the United States, located near the edge of the known geographic range of some species of Tridacninae, leading to expectations of decreased abundances and greater population vulnerability (Hidas, Ayre & Minchinton, 2010; Neo et al., 2017; Masanja et al., 2023). Previous surveys have identified the presence of Tridacna maxima, T. squamosa, and historical records have found shell remnants of Hippopus hippopus. T. derasa was introduced for aquaculture in the late 1980’s, but are no longer present within territorial waters (Ponwith, 1990; Green & Craig, 1999; Linnekin et al., 2006).

The coastline of American Sāmoa falls under various levels of jurisdiction and protection status, the majority under the resource management of the local American Samoan Government (ASG) Department of Marine and Wildlife Resources (DMWR), with supported enforcement staff from Marine Patrol. The enforcement of fisheries regulations in American Sāmoa is constrained by limited governmental resources, including insufficient vessel availability and staffing shortages (Levine & Richmond, 2014). Fa‘asao (village-based fishery closures) represent a traditional form of Indigenous management recently formalized to enhance local resource stewardship. Each village establishes its own terms and commitment periods for fishery restrictions, which in many cases have been integrated into the Community-based Fisheries Management Program (CFMP) (Levine & Allen, 2009). The CFMP serves a dual purpose: it strengthens the capacity of village communities to manage and protect nearshore resources with government support, while simultaneously expanding the enforcement and surveillance capabilities of the Department of Marine and Wildlife Resources (DMWR) (King & Faasili, 1999; Fa’asili & Sauafea, 2001). This collaborative approach helps mitigate limited capacity for fisheries regulation enforcement by the American Samoan government, which is constrained by restricted vessel access and staffing resources. Additionally, there are federal agencies, such as US Fish and Wildlife Service (USFWS) and NOAA National Marine Sanctuary in American Sāmoa (NMSAS), which each have established no take zones within the territorial waters that have been present for over 40 years (Wegman & Holzwarth, 2006; Raynal, Levine & Comeros-Raynal, 2016). Likewise, the National Park of American Sāmoa restricts take to subsistence fishing only within park boundaries (National Park of American Samoa (NPSA), 2023).

Dispersal potential of faisua (giant clams), across the Samoan Archipelago are not yet fully understood, but given the proximity of neighboring islands and the speed of the South Equatorial Current (SEC), it is plausible that larvae are dispersed across the region (Kendall & Poti, 2011). Muliāva (Rose Atoll National Marine Monument), is a remote, uninhabited atoll that is fully protected as a Marine National Monument. It has been identified as a potential population refuge for giant clams, harboring an estimated 27,800 clams during a 1994 survey (Radtke, 1985; Green & Craig, 1999). The SEC flows westward from Muliāva towards the rest of the archipelago at a speed of approximately 1 km/hour (Kendall & Poti, 2011), making larval dispersal from Muliāva to the nearest islands, 138 km away, likely given a larval duration of 10–14 days for these species (Ellis, 1997). However, empirical studies of multispecies connectivity across similar distances often challenge assumptions about dispersal potential based simply on proximity and currents, so additional work is needed to confirm dispersal patterns among sites within the archipelago (Weersing & Toonen, 2009; Toonen et al., 2011; Crandall et al., 2019).

This population status assessment is a first step in evaluating the population trend dynamics and extinction risk of giant clam species within the Samoan Archipelago. Considering the pending ESA listing petition and the recent 2024 status review conducted by the National Marine Fisheries Service, which identified overexploitation and inadequate regulatory mechanisms as primary threats to several Tridacninae species, this research is critical. By providing updated, region-specific data on the distribution, abundance, and potential misidentification of species such as T. noae, this study addresses key gaps identified in the petition and status review. The findings will inform conservation strategies, enhance resource management, and contribute to the broader understanding of giant clam population dynamics. Ultimately such data are needed to support efforts to mitigate extinction risks and promote sustainable management of these ecologically significant species.

Materials & Methods

To remain consistent with previous historical surveys, all methods were based on Green & Craig (1999), which consisted of archipelago wide surveys across American Sāmoa. Abundance was estimated via SCUBA surveys on three 50 m × 2 m belt transects at 10 m depth for each site across the archipelago (see Fig. 1). Surveys were conducted on Tutuila, Manuʻa Islands (Ofu, Olosega, and Taʻu). Due to remote diving restrictions, shallow water (<2.5 m depth) snorkel surveys were conducted at Muliāva (Rose Atoll). Additional sites were added around Tutuila, Ofu, Olosega, and Taʻū, to improve geographically survey coverage and island representation.

Surveys were conducted during four expeditions occurring between 2022 and 2024 (see Table 1). Sites were surveyed across multiple years because of the spatial extent, vessel access, or inaccessibility due to weather conditions within a single expedition. We completed a total of 194 transects, covering 2.32 hectares across 88 sites, among six islands of the archipelago. For the purpose of this research, sites were classified into the following levels of marine management: federal no take (NMSAS); subsistence & remote harvest (NPSA waters); remote, inaccessible (Muliāva); village protected areas; or none (Existing Government Regulations) (see Table S1). Remoteness was categorized by limited road access and increased distance from population center.

Figure 1 Map of American Sāmoa survey sites.

Location of (A) each island in American Sāmoa and (B–D) topographic maps of each island including coral reefs, survey sites at 10 m depth, and urban areas represented by color shading. Sites in red indicate historical survey locations, green indicate additional surveys in this study. Figure adapted from Green et al. (2022). For more details on studied locations, see Table S1.

Population densities were categorized based on Neo et al. (2017) into three classes of abundance: Abundant (100–10,000/ha); Frequent (1–10/ha) and Rare (<0.1/ha). For each clam found on our transects, we recorded: size, depth, species, valve margins, photo documentation, and notes about morphology (intake siphon tentacles, papillae, teardrop shape, etc.). Following Green & Craig (1999), clams were assigned to one of three size classes based on shell length measured in a straight line from tip to tip along the longest axis of the shell: Recruit < five cm, Immature 5 < 12, and Mature ≥ 12 cm. These size categories are consistent with best fit functions of shell size by gonadosomatic index of Tridacna maxima in French Polynesia (Menoud et al., 2016).

It is noteworthy that the surveys conducted in 2018 in Manuʻa islands (Ofu, Olosega, and Taʻū) were impacted by weather conditions, and clam abundances were likely underestimated due to difficulty surveying and limited survey sites (A Green, pers. comm., 2025). Due to the incomplete surveys in 2018, we also ran comparative analyses of the time series, one with all years included, and another omitting the data from 2018.

Table 1 Timeline of scuba surveys and the respective geographic survey scales.

Time period	Survey locations		
1994/1995	Main islands (Tutuila, Aunuʻu, Ofu, Olosega and Tau) and two remote atolls (Muliāva and Swains).	Green & Craig (1999)	
2002	Tutuila, Aunuʻu, Ofu, Olosega and Tau	Green (2002)	
2018*	Tutuila, Aunuʻu, Ofu, Olosega, Tau and Rose Atoll	Green et al. (2022)	
2017–2018	National Park, Tutuila District	This study	
2022	Ofu, Olosega, Taʻu and Muliāva. National Park, Tutuila District	This study	
2023	Tutuila Wide Surveys	This study	
2024	Tutuila Wide Surveys	This study	
Notes.

* Due to severe weather, limited sites were surveyed on Ofu, Olosega, and Ta’ū in 2018.

Statistical analysis

We conducted a series of univariate and factorial ANOVAs to assess spatial and temporal variation in giant clam density and size across American Sāmoa. Analyses tested the effects of island, year, protection status, and their interactions. Post hoc Tukey HSD tests were used to identify pairwise differences, and eta-squared values were calculated to estimate effect sizes. Eta-squared values (η2) provide a measure of effect size for a specific variable in an ANOVA model after accounting for the other variables. See Table S4 for detailed results, or github repository https://github.com/Pgjhmb/Giant-Clam-Population-Trends-in-American-Samoa for complete R code. All statistical analyses were performed in RStudio (v2023.03.0+386), using base R functions from the following packages (functions): stats (aov, TukeyHSD), rstatix (wilcox_test, shapiro_test, kruskal_test), effectsize (eta_squared) (R. Development Core Team, 2008; Ben-Shachar, Lüdecke & Makowski, 2020; Kassambara, 2023). Data were visualized using ggplot2.

Muliāva methodology

Due to equipment, weather and compliance constraints, we were unable to conduct SCUBA surveys on Muliāva, and instead conducted shallow-water (<2.5 m depth) snorkel surveys. Survey sites were located on shallow coral clusters (bommies) and along the tops of deeper pinnacles in the interior lagoon, not on the reef slopes. Abundances were estimated based on the surface area covered by the survey. In the case of coral bommies, a surface area estimation was done using a cylindrical measurement of the bommie or pinnacle. Each clam on survey was photo-documented and morphometrics were measured (depth, shell length, valve margins) and descriptions were noted (color, scute shape/size, siphon tentacle complexity). Observed clams were identified based on morphology in the field.

Results

Temporal trends in giant clam populations across American Sāmoa

Due to inclement weather, we were unable to survey historically high abundance sites, so the 2018 time series datapoint was not included in our analysis. Supplemental Data was provided from territorial reports, compiled by the CRAG and DMWR (Green et al., 2022).

Comparing data from all survey years reveals clam densities strongly linked to island (F(4,118) = 17.47, p < 0.01,η2= 0.37), but not time (Fig. 2, Table 2). Even including 2018, when severe weather prevented surveys at the historically highest density sites on Ofu, Olosega, and Taʻū, there was no significant relationship between clam abundance and year (F(3,119) = 1.703, p > 0.05,η2= 0.04). Tutuila measured an average clam density of 83.5 in 2022–2024, with specific sites showing a density of over 400 clams per hectare.

Figure 2 Mean density (clams/ha) of live giant clams surveyed on reef slopes (10 m) on six islands in American Sāmoa, in survey years 94/95, 2002, 2018, and 2022–24 (Green & Craig (1999), Green (2002), Green et al. (2022) and this study).

On reef slope surveys, the highest mean density was observed on Taʻū (n = 21), with 811.89 clams per hectare (SE = 288.6), followed by Aunuʻu (n = 3) at 300 clams per hectare. Ofu (n = 18) had a mean density of 138.57 clams per hectare (SE = 72.66), while Olosega (n = 6) exhibited a mean density of 116.85 clams per hectare (SE = 83.15). In contrast, Tutuila (n = 114) exhibited the lowest mean density at 83.47 clams per hectare (SE = 18) (Fig. 2).

Impact of protection levels on clam populations

We investigated differences among sites, starting with broad categorizations of sites where take is allowed compared to those that are fully protected (Fig. 3). Using the most recent survey data (2022–2024), we compared Open Access sites, where fishing is allowed in various forms, to Marine Managed Areas, where all forms of fishing are banned (Federal No Take, Village Protected, and Inaccessible sites) (Fig. 3). Clam density varied widely among Open Access sites (n = 32, mean = 87.4, SE = 21.2) (n = 138 transects, mean = 207 clams/ha, SE = 44.7) but did not differ significantly (F(1,121) = 0.655, p 0.42, η2= 0.005), from Marine Managed Areas (n = 53 transects, mean = 958 clams/ha, SE = 468). However, this comparison confounds island with level of protection because only one type of site exists on most islands. Tutuila is the only island with both Marine Managed Areas and Open Access sites (Fig. 3). To further investigate how management impacts clam density, these two protection categories were further partitioned into various forms of marine management strategies (Federal No Take, Village Protected, remote, subsistence and remote, none, and Inaccessible sites) for further analysis (Fig. 4).

Table 2 Clam densities across surveyed timepoints along the American Samoan islands, with standard deviation and standard error.

 	1994	2002	2018	2022–2024	
 	Mean	SD	SE	Mean	SD	SE	Mean	SD	SE	Mean	SD	SE	
Aunuʻu	80	–	–	100	–	–	67	–	–	300	–	–	
Tutuila	14.1	26.23	6.6	119.6	31.3	7.8	41.2	89.4	22.3	83.5	112.2	18.2	
Ofu	260.0	311.13	220	266.7	236.9	167.5	50.0	70.7	50.0	138.6	178	72.7	
Olosega	230.0	183.85	130	420.0	141.4	100.0	50.0	70.7	50.0	116.9	117.6	83.2	
Tau	305.0	305.0	152.5	860.0	577.1	288.6	350.0	23.6	11.8	811.9	288.6	109.1	

Figure 3 Mean density (clams/ha) of live giant clams surveyed on six islands in American Sāmoa from 2022–2024, delineating protection into alternative marine managed areas and open access areas.

Alternative marine managed areas (Federal No Take, Village Protected, and Inaccessible) and Open Access (Existing governance, Remote, Subsistence & Remote).

Figure 4 Mean density of live giant clams across islands, levels of protection, and corresponding clam sizes.

(A) Mean density (clams/ha) of live giant clams surveyed on reef slopes (10 m) on five islands in American Sāmoa delineated by level of protection, relative to shallow snorkel surveys on Muliāva during 2022–2024 surveys. (B) Mean density of live clams across Tutuila delineated by level of protection, with abundance classes as defined by: Abundant (100–10,000/ha); Frequent (1–10/ha) and Rare (<0.1/ha). Numbers above bars indicates count of transects per level of protection. (C) Live clam size (cm) across levels of protection. Numbers above bars indicates count of clams measured. (significance: *** = 0.001, ** = 0.01, * = 0.05, all other relationships are not significant).

Throughout the 30 years of surveys (Fig. 2), giant clam densities did not change significantly over time (F(3,119) = 1.703, p > 0.1,η2= 0.04), but showed consistent large variation among islands (F(4,118) = 17.47, p < 0.001,η2= 0.37) and across levels of protection (F(5,117) = 3.206, p < 0.01,η2= 0.12).

Remote areas (n = 27 transects) had relatively high clam densities, averaging 461.3 clams per hectare (SE = 138.32), whereas Subsistence & Remote zones within National Parks (n = 33 transects) had a lower average density of 207 clams per hectare (SE = 42.54). Federal no-take zones (n = 9 transects) exhibited an average density of 100 clams per hectare (SE = 100), while Village Protected areas (n = 19 transects) had a mean density of 90.0 clams per hectare (SE = 12.7). Areas under Local Government Regulations (n = 72 transects) showed an average density of 119 clams per hectare (SE = 57). Muliāva, classified as Inaccessible, had the highest recorded clam densities due to its uninhabited status, averaging 1,166.39 clams per hectare (n = 34 transects, SE = 573.98).

Because many islands contain only one or a few types of management regime, we also compare among surveys (n = 99 transects) on Tutuila, the only island on which all management types exist (Figs. 4B & 4C). Among these Tutuila management zones, clam densities differed significantly based on protection level (F(4,33) = 10.3, p < 0.001,η2= 0.56; Fig. 4B). Among management regimes, Subsistence & Remote sites had the highest densities of giant clams, followed by Remote and Village Protected, whereas the Federal no-take sites held the lowest mean density of clams overall on the island of Tutuila (Fig. 4B).

Size distribution

Within the 2022–2024 surveys, Tridacna size varied significantly across both levels of protection (p < 0.001,η2= 0.14), and across islands (p < 0.001,η2= 0.14) (see Fig. 5, Table 3).

Figure 5 Live giant clam sizes of across islands in American Sāmoa found in 2022–2024 surveys.

Giant clam size in violin plot and box plot across islands in American Sāmoa found in 2022–2024 surveys, plotted on the primary Y axis to the left. The red tick marks on the primary Y axis indicate the breaks between size classes. Purple circles indicate outliers and the number of clams measured (n) is listed at the top of figure. Shaded line in the background indicates mean giant clam density (number/ha) across islands plotted relative to the secondary Y axis on the right.

Incorporating historical surveys, clam sizes varied significantly between islands (p < 0.001,η2= 0.18) and across years (p < 0.001,η2= 0.07); Fig. 6, Table 4). Muliāva (n = 58) and Aunuʻu (n = 37) had the largest average clam sizes, measuring 14.2 cm (SE = 0.428) and 13.35 cm (SE = 1.519), respectively. In contrast, Taʻū (n = 40) exhibited smaller average clam sizes at 8.79 cm (SE = 0.390). Tutuila (n = 85) displayed a wide range of sizes, with an overall average of 12.21 cm (SE = 0.733).

Figure 5 further highlights substantial differences in size class distributions between islands. Tutuila had moderate population densities across size classes, averaging 82.06 clams per hectare. However, in the Manuʻa islands, where population densities are much higher (116–718 clams per hectare), there were minimal recruits on Ofu and Olosega islands.

Species composition

This was the first population survey that distinguished between cryptic species in American Sāmoa since the discovery of T. noae (Marra-Biggs et al., 2022) (Fig. 7). The vast majority of individuals (96.7%) in this survey were identified as T. maxima. Only 14 T. noae and nine T. squamosa were recorded across all transects, with 10 of the T. noae and all T. squamosa observed on the island of Tutuila.

Table 3 Giant clam density (mean/ha) across islands during 2022–2024 surveys, delineated by size classes, with standard deviation and standard error.

Classes are categorized as (Recruit ≥ 5 cm, Juvenile 6 ≥ 12 cm, and Mature ≤ 12 cm) as defined by Green & Craig (1999), Menoud et al. (2016) and Green et al. (2022).

		Density	St Dev	St Error	
	Survey
Count	Recruit	Juvenile	Mature	Total	Recruit	Juvenile	Mature	Recruit	Juvenile	Mature	
Tutuila	114	12.61	25.23	36.94	82.06	24.0	58.5	50.2	3.9	9.5	8.1	
Aunuʻu	3	100.00	0.00	200.00	300.00	–	–	–	–	–	–	
Ofu	18	22.22	11.11	88.89	139.57	40.4	27.2	75.0	16.5	11.1	30.6	
Olosega	6	16.67	16.67	83.33	116.85	23.6	23.6	70.7	16.7	16.7	50.0	
Tau	21	242.86	238.10	280.95	811.89	135.7	138.0	141.2	51.3	52.2	53.4	
Muliāva	102	41.65	220.90	843.37	116.39	232.1	812.3	2,588.1	39.8	139.3	443.9	

Figure 6 Mean sizes across reef slope surveys (10 m depth) along the different islands (Green & Craig, 1999; Green, 2002; Green et al., 2022, this study).

Table 4 Giant clam size (cm) on reef slopes (10m) with standard deviation and standard error in 1994–2024 surveys.

 	Mean Size	N	SD	SE	
 	1994/95	2002	2018	2022/24	1994/95	2002	2018	2022/24	1994/95	2002	2018	2022/24	1994/95	2002	2018	2022/24	
Tutuila	14.08	17.98	18.89	11.56	12	24	22	80	10.16	8.71	4.45	6.40	2.93	1.78	0.95	0.72	
Aunuʻu	10.00	19.17	22.25	11.78	4	3	2	9	6.78	5.39	15.20	7.14	3.39	3.11	10.75	2.38	
Ofu	8.35	13.11	15.33	13.35	26	35	3	20	7.29	5.64	2.52	6.79	1.43	0.95	1.45	1.52	
Olosega	17.09	13.42	17.67	12.43	23	42	3	7	7.01	6.11	12.50	5.00	1.46	0.94	7.22	1.89	
Taʻū	6.89	9.29	12.65	8.79	92	311	21	165	4.03	3.75	3.15	5.01	0.42	0.21	0.69	0.39	

Figure 7 Extant clam species in American Sāmoa Tridacna noae (A), Tridacna squamosa (B), and Tridacna maxima (C & D).

(A) Tridacna noae characterized by teardrop-shaped mantle spots encircled with a golden ring and notably lacking a continuous row of hyaline organs (eyes). (B) Tridacna squamosa distinguished by irregular blotches and its branching, elaborate guard tentacles. (C–D) The most common species of giant clam in American Sāmoa, Tridacna maxima exhibits wide variation in mantle coloration; however, the presence of a peripheral row of eyes is a consistent feature. (C) A color variant T. maxima and (D) a common blue color morph from Tutuila illustrate this diagnostic eye band while lacking the ringed patterning. All images were photographed in Tutuila, American Sāmoa. Photo credit: Paolo Marra-Biggs.

Muliā va shallow population densities

Historical surveys of Muliāva (Green & Craig, 1999), reported clam densities ranging from 300 up to 11,600 clams per hectare on the shallow pinnacles. Although conducted with a different methodology using shallow snorkel transects here, Muliāva still had the highest population of all the surveyed islands with a mean population density of 1,166.39 clams per hectare (n = 34, SE = 573.98). The mean density on shallow pinnacles was 167.5 clams per hectare (SE = 35.6). Instead, we found coral bommies were the most populated habitat in our surveys, with an average density of 1,729 clams per hectare (SE = 245.3), though showing large variation (range: 0–18,734.9 clams per hectare). Despite comprising only 0.9% of the total area we surveyed, coral bommies (n = 22) accounted for 95% of all individuals due to their high clam densities. In contrast, the reef flats (n = 3), which made up a larger proportion of surveyed habitats across all locations, had reduced clam populations with an average of 33 clams per hectare (SE = 12.4) and a maximum of 100 clams per hectare.

Discussion

Giant clam populations are decreasing throughout much of their native ranges due to overharvesting and habitat destruction (Meadows, 2016; Dolorosa et al., 2024), reinforcing the need for consistent monitoring protocols to assess the rate of decline. American Sāmoa is no exception to these anthropogenic pressures, and limited territorial resources has resulted in large spatial and temporal data gaps. This study replicated historical surveys to assess the state of current giant clam populations, and to provide management agencies data needed for mitigative actions. To inform management agencies, these survey data identified areas with high and stable abundances, highlighting how different marine resource protection strategies can affect clam populations and best maintain clam stocks under territorial resource constraints.

Archipelago wide population trend

In this study, we followed the three abundance categories established by Neo et al. (2017) for giant clams. Because American Sāmoa lies near the edge of the Tridacninae native geographic range, natural abundances are expected to be low (Hidas, Ayre & Minchinton, 2010; Masanja et al., 2023). Our results are consistent with those of Neo et al. (2017) that highest clam densities were found within areas remote from human populations that have low historical fishing pressure. However, surveys in populated regions within close geographic proximity have much higher variability, indicating multiple factors are influencing giant clam densities, and a deeper investigation into these population drivers would be beneficial to understand underlying causes. Despite the perception of a precipitous decline, regions with high giant clam abundance persist, and average population densities range from 83.5 to 1,111 clams per hectare across the islands. However, many sites close to high human populations on Tutuila lacked any living clams on transects, likely due to heavy fishing pressure and increased land-based pollution (Neo et al., 2017; Comeros-Raynal et al., 2019; Comeros-Raynal et al., 2021; Shuler & Comeros-Raynal, 2020; Houk et al., 2020). Despite Tutuila hosting 98.3% of the human population and increased fishing pressure, the island still contained high-abundance areas which elevated the average density to 83.47 clams per hectare, classifying it as “frequent” clam abundance overall (Neo et al., 2017). However, as discussed earlier, giant clams are susceptible to reduced reproductive success at low population densities, and therefore identifying regions of high population density is valuable for designating priority sites for conservation and restoration efforts (Parsons, 2010). Patchy distributions could also be caused in part by the island geography and availability of clam habitat. For example, Tutuila has more large bays than other islands. Embayments can have higher retention and larval residence time that can lead to self-recruitment (e.g., Coleman et al., 2023) and potentially contribute to higher local clam densities.

Weather conditions limited 2018 surveys around the Manuʻa islands, preventing surveys at multiple sites on northern Taʻū, Ofu and Olosega, which include some of the historically highest density sites for giant clams. By omitting those sites, the island-wide mean densities reduced in 2018 (Fig. 2). Despite such differences in the number and location of surveys, the clam densities (Fig. 2) and size classes (Fig. 6) remain relatively stable over time relative to historical baselines. Thus the conservation concern appears less dire in American Sāmoa than previously believed. It is important to note, however, that these mean values per island come from averaging transects across sites with no living clams in highly populated areas and those with abundant clams under Fa‘asao village management. This extreme variation in clam density highlights not only differences in abundance and size among locations shown in Fig. 4, but also the importance of effective management in maintaining viable reproductive clam populations.

Size class distribution

Across 111 measured clams around Tutuila during the 2022–2024 surveys, the average size was 11.56 cm, near the estimated size of maturity at terminal sexual phase (Copland & Lucas, 1988). However, when partitioning into levels of protection, both existing government regulations and village-protected areas measured average sizes of 17 cm, higher than reported sizes in Federal no take areas across Tutuila and Aunuʻu (11.2 cm). Based on size and density data, remoteness and village enforcement stand out as best for giant clams in American Sāmoa but being remote is not a viable management action that can be implemented by resource managers. Thus, village enforcement appears to be the most effective of available management strategies, maintaining both high abundances (Fig. 4B) and higher average clam sizes (Fig. 4C).

In contrast, the low number of clams in the smallest size classes in the Manuʻa Islands (particularly Ofu & Olesega, Fig. 4), coupled with higher densities, suggests low fertilization success or recruitment, making these populations more vulnerable to decline (Caley et al., 1996). However, Taʻū demonstrates a stable and thriving population, with a large representation of clams across juvenile and mature size classes and high density. One hypothesis to explain the strong recruitment class and abundant juvenile clams at Taʻū (Fig. 5) is larval supply from dense populations of clams on Muliāva.

The size frequency distributions show evidence for limited recent recruitment at many sites (Fig. 5). Comparative rarity of the smallest size classes on Aunuʻu, Ofu & Olesega compared to Taʻū implies limited recruitment and decreased potential for rebound from catastrophic population decline. Ofu and Olesega have maintained relatively consistent mean clam sizes through time, but both Aunuʻu and Tutuila show a steep decrease in mean clam sizes since 2018 (Fig. 6), which raises concern for future population trends on these islands if harvest of adults continues or increases.

Relative species abundances

Despite the recent discovery of Tridacna noae, T. maxima was overwhelmingly dominant among clams observed in this study, comprising 96.7% of the 453 clams we surveyed. T. noae was found mostly in village-managed areas, where they were observed most commonly in shallow habitats less than 3 m—zones that would otherwise be highly vulnerable to harvest. This finding further supports the value of fine-scale habitat protection and the effectiveness of localized monitoring and enforcement. The relative scarcity of T. noae lends confidence that previous surveys of giant clams, which assumed all clams were T. maxima, remain informative. However species composition data should be integrated into long-term monitoring plans to detect if relative species abundances change through time, and to further understand functional diversity and adaptive potential among giant clam populations in American Sāmoa.

Level of protection

We examined data from sites under different management strategies to gain insight as to whether no-take reserves currently benefit giant clam populations. The fact that most islands have a limited array of management regimes complicates comparisons because differences exist among islands in the amount and suitability of clam habitat. Further, human population density along the coastline almost certainly has a strong influence on clam abundance. In the case of American Sāmoa, marine reserves that are managed by the adjacent villages outperformed other types of enforcement strategies. Particularly on main island of Tutuila (Fig. 3), where 98% of humans in American Sāmoa live (US Census Bureau, 2020), subsistence and remote areas where few people can reach significantly outperform the other forms of protection (Fig. 4B), followed most closely by village-enforced fishing areas. However, managers cannot alter proximity of sites to human populations which is why we classified remote sites as their own group for these comparisons. In Fig. 4A, it may appear that Muliāva (which is entirely inaccessible due to its remoteness and is also classified under Federal No Take status) and Aunuʻu (on which surveys were only located within Federal No Take areas), perform best as management strategies. However, this outcome conflates variation among islands with management strategy because the two overlap entirely at these sites. When all islands are grouped together and analyzed by management strategy, remote sites outperform all other types of protection, as in the case of Muliāva (Fig. 4C). Likewise, subsistence & remote sites were found to contain significantly more giant clams (mean density of 207.17 clams per hectare) compared to the remaining forms of protection (Remote = 108.33, SE = 34.4), with Village Protected following closely behind (Mean = 93.8, SE = 20.4). Thus, of the possible management strategies available to natural resource managers, village protection outperformed both the local government agency laws (“none”), and Federal no-take MPAs.

Focusing only on the main island of Tutuila (Figs. 4B, 4C) which is the only island on which all management regimes exist, the level of protection under Fa‘asao village management shows that both the giant clam density and size class ranges are higher than areas under other forms of protection. Villages that enforce traditional methods of resource protection typically enforce from within the local population of residents. This not only increases buy-in of the local population, but also allows for multiple residents to act as lookouts for potential poaching and provide early warning systems to prevent harvest (Atlas et al., 2021). Traditional village-based marine protection—often relying on direct community oversight, Indigenous cultural standards, and social incentives—is highly effective, particularly in regions like the Central and South Pacific and Southeast Asia due to their deep local and traditional knowledge and cultural practices (Pajaro, 2010; Thornton & Scheer, 2012; Winter et al., 2020). Village managed areas often have higher compliance rates, more effective monitoring, and cultural stewardship (Léopold et al., 2013; Aswani, Albert & Love, 2017; Parker, 2021). In contrast, the abundance of clams in Marine Managed Areas fell within the lowest category of rare (Neo et al., 2017), despite being protected at the highest level of federal no-take management on Tutuila. This area is within the jurisdiction of the National Marine Sanctuary of Fagatele Bay, where coral health has persisted and improved within the last 40 years, however “biomass of fisheries species in Fagatele Bay remains low compared with many other sites throughout the Territory” (Green et al., 2022). Likely, no-take MPAs with insufficient enforcement provide little protection if the local community disagrees with the management approach, or if they are accessible places where poaching can occur without severe consequences.

Conclusions

Our findings reinforce that community-based management approaches can effectively sustain clam populations and may serve as a more adaptive and culturally sensitive alternative to top-down enforcement under ESA. To further improve conservation outcomes, giant clam monitoring should be incorporated into the existing annual survey programs, providing a finer resolution of temporal trends in the Samoan archipelago. Additionally, since current assessments are limited to a single depth contour (10 m), a more comprehensive survey across depth gradients is recommended to identify potential depth refugia, assess habitat limitations, and improve our understanding of species-specific ecological affinities in the archipelago. Such efforts require hosting workshops to help local fishers correctly identify cryptic species (T. noae and T. squamosa), particularly given their detection primarily in village-managed areas to date. Further education and community training is needed for the reliable monitoring of species-specific populations and to better inform species-specific management plans. Frequent, spatially distributed monitoring—especially when paired with community engagement—will be key to ensuring long-term persistence of giant clam populations under changing environmental conditions.

Population densities of giant clams are variable across American Sāmoa, with highest densities concentrated in a limited number of remote and village-protected areas. Although global concern over giant clam declines has prompted consideration for endangered species listing, our findings suggest that such designation does not appear to be the best management pathway for American Sāmoa where ESA status would equate to federal no-take management and would likely limit village management efforts. The relative stability of clam densities relative to historical baselines, alongside clear evidence that village-enforced protection currently outperforms other management strategies, highlights the value of strengthening local stewardship rather than relying solely on federal regulations to conserve these iconic coral reef species.

Supplemental Information

Supplemental Information 1 Site locations and level of protection

Survey site names have been removed and GPS coordinates are limited to one decimal place to protect sensitive giant clam habitats from overharvesting. Additional supporting data are available upon reasonable request from the authors, but full access to site-specific information is restricted due to legal and conservation concerns. Table adapted from Green & Craig (1999) and Green et al. (2022).

Supplemental Information 2 Mean density (clams/ha) of live giant clams surveyed on reef slopes (10 m) on six islands in American Sāmoa by Green et al. in 1994/95, 2002, 2018, and 2022 in this study

Supplemental Information 3 A) Mean density (clams/ha) of live giant clams surveyed on reef slopes on six islands in American Sāmoa (2022-2024) delineated by type of protection

B) Mean density (clams/ha) of live giant clams surveyed on reef slopes (10 m) on Tutuila, American Sāmoa (2022-2024) delineated by type of protection.

Supplemental Information 4 Ad Hoc Analyses of Giant Clam Surveys

Various statistical results from our R code, using datasets variety of datasets (All years, All years except 2018, and 2022-2024). All code was run using RStudio (v2023.03.0+386).

Supplemental Information 5 Under Over photo of Tridacna noae in Ofu, American Sāmoa

Photo Credit: Paolo Marra-Biggs

Supplemental Information 6 Tridacna noae on reef shelf in Muliāva, American Sāmoa

Photo Credit: Paolo Marra-Biggs

Supplemental Information 7 Tridacna maxima in on a reef survey in American Sāmoa

Photo Credit: Paolo Marra-Biggs

Supplemental Information 8 Giant clam densities and sizes across surveys in American Samoa

Supplemental Information 9 Giant clam size ridgeline plot across islands within the Samoan Archipelago

Fa‘afetai tele lava to the U.S. Fish and Wildlife Service and other organizations for their invaluable logistical support and scientific guidance. I also extend sincere thanks to the Hawaiʻi Institute of Marine Biology and the members of the ToBo Lab for their expertise and collaboration.

This project is the culmination of a collaborative effort among numerous federal, territorial, and university partners who worked together to conduct a comprehensive assessment of giant clam populations across the American Samoan Archipelago. For their scientific guidance and assistance with giant clam surveys, we would like to express our deepest gratitude to Dr. Mareike Sudek, Michael Marsik, Michele Felberg, Jeremy Raynal, Motusaga Vaeoso, Georgia Coward, Tim Clark, Fuiava Bert Fuiava, Daniel George, Scott Burch, Victoria Barker, Amélie Tagliaferro, and Johann Vollrath.

Additional Information and Declarations

Competing Interests

Author Contributions

Data Availability

Robert Toonen is an Academic Editor for PeerJ.

Paolo Marra-Biggs conceived and designed the experiments, performed the experiments, analyzed the data, prepared figures and/or tables, authored or reviewed drafts of the article, and approved the final draft.

Eric K. Brown conceived and designed the experiments, performed the experiments, analyzed the data, prepared figures and/or tables, authored or reviewed drafts of the article, and approved the final draft.

Domingo Galgo Ochavillo conceived and designed the experiments, analyzed the data, prepared figures and/or tables, authored or reviewed drafts of the article, and approved the final draft.

Alison L. Green conceived and designed the experiments, analyzed the data, prepared figures and/or tables, authored or reviewed drafts of the article, and approved the final draft.

Alice Lawrence conceived and designed the experiments, performed the experiments, authored or reviewed drafts of the article, and approved the final draft.

Carlos Tramonte performed the experiments, authored or reviewed drafts of the article, and approved the final draft.

Valentine Vaeoso conceived and designed the experiments, performed the experiments, authored or reviewed drafts of the article, and approved the final draft.

Ian Moffitt conceived and designed the experiments, performed the experiments, authored or reviewed drafts of the article, and approved the final draft.

Kersten Schnurle conceived and designed the experiments, performed the experiments, authored or reviewed drafts of the article, and approved the final draft.

Nury Molina performed the experiments, authored or reviewed drafts of the article, and approved the final draft.

Robert J. Toonen conceived and designed the experiments, analyzed the data, prepared figures and/or tables, authored or reviewed drafts of the article, and approved the final draft.

The following information was supplied regarding data availability:

The R Code is available at GitHub and Zenodo:

- https://github.com/Pgjhmb/Giant-Clam-Population-Trends-in-American-Samoa

- Marra-Biggs, P. (2025). Status and trends of giant clam populations demonstrate the effectiveness of village-based protection in American Sāmoa. Zenodo. https://doi.org/10.5281/zenodo.15413140

The dataset is also available in the Supplementary Files.

The raw data is available at Zenodo:

- Marra-Biggs, P. (2025). Datasets used in the manuscript titled ”Status and trends of giant clam populations demonstrate the effectiveness of village-based protection in American Sāmoa” [Data set]. Zenodo. https://doi.org/10.5281/zenodo.17282668

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
