# Peer review of "Status and trends of giant clam populations demonstrate the effectiveness of village-based protection in American Sāmoa"

_PeerJ, doi:10.7717/peerj.20290_

## Round 0.1 · original submission · Major Revisions

· Academic Editor

Major Revisions

Dear Dr. Marra-Biggs,

You can find the comments and suggestions of the expert reviewers in the attached reports. As you will see, expert reviewers have pointed out the critical errors. Consequently, a major revision is needed for your article.

I request that you improve your manuscript following the reviewers' suggestions

Sincerely

Reviewer 1 ·

Basic reporting

This study addresses a critical knowledge gap in the population status of giant clams in American Samoa—a region with limited recent data despite the ecological and conservation significance of these species. I commend the authors for their field efforts in this remote location. However, the manuscript, in its current form, requires major revision due to several structural, methodological, and clarity issues.

The introduction is overly long and lacks structure. It should be reorganized to:
• First, state the species diversity and known historical presence in American Samoa.
• Then, update readers on global and local conservation status (IUCN and ESA).
• Follow with a brief section on cultural and economic importance (if any).
• Too much emphasis is placed on reproductive constraints and connectivity, which are more appropriate in the discussion.
• Cryptic species discussion is premature without first establishing baseline species diversity.
Suggested restructure:
1. Background and species richness in American Samoa.
2. Global and national conservation status.
3. Cultural/economic relevance.
4. Rationale for current study (e.g., confusion from past surveys, cryptic diversity).
5. Protection categories and management relevance.

Experimental design

This study is important to fill the knowledge gap on giant clam status in American Samoa and will be significant for the ongoing work in updating their population status globally. However, numerous issues compound any meaningful interpretation of results.
• Lack of a clear description of statistical design and approach.
In fact, there is no description of a statistical approach at all in the Materials & Methods section. The author should describe what are the statistical approach they employed and on what platform they performed such analysis.
• Check for consistency in the unit of measurement
Throughout the manuscript, especially in the Figure caption, the label in the Figure, and the text. There are multiple inconsistencies in the unit of measurement used (regarding the density of the clam). Please refer to the attachment for line-to-line comments.
• Formatting and citation format
There are a lot of inconsistencies in formatting, such as paragraph space, units, and citation format. Please refer to the attachment for line-to-line comments.

Validity of the findings

Hard to assess at this stage with a critical flaw in the statistical approach and a lack of well-defined research questions.

Additional comments

Line-by-line comments by sections:
Abstract:

L38-39: “considerable variation among villages”?

L42: What is the justification for crypticity in T. squamosa? Cryptic diversity is not common for T. squamosa, especially those that closely resemble are all endemic to the Indian Ocean.

Introduction:
L58-74: This paragraph can be further concise into at most three sentences.

L76-93: This paragraph is about their global status. Both paragraphs can be combined, and please provide the newest update of the IUCN status for the species. This can be followed by the ESA under American regulations.

L95-128: These two paragraphs are about reproductive constraints leading to the Allee effect. It can be combined and concise into one paragraph. Some information can also be considered for migration to the discussion.

L130-143: This paragraph is on cryptic species. When there is no mention of actual or historical species diversity records in the previous paragraph, discussion on cryptic diversity is irrelevant and out of context. This can be combined with the subsequent paragraph.

L141: typo with “Tridacna Hippopus hippopus”? And please italicize when mentioning
species.

L145-158: This paragraph is about the population status of American Samoa. This is the utmost crucial information in the whole manuscript and should have been mentioned after the first paragraph. And here should mention how many giant clam species are actually recorded. And continue to elaborate on cryptic diversity that could confound the previous survey (hence, emphasize why this study is so important)

L160: This paragraph is about the management and regulation related to giant clams in American Samoa. In the data analysis, there is an intention to differentiate different protection levels (no-take, subsistence, and so on). Here, the authors should actually elaborate why it is important to look at different protection levels and clearly define those protection categories.

L182: On the dispersal potential of the giant clam. “Faisua” is referring to the local vernacular for giant clam, which is an important information to highlight. Authors should spend at least one or two paragraphs introducing the cultural and economic importance (if any) of the giant clam in American Samoa. The connectivity of giant clams in the region is less important and would be more suitable in the discussion section.

Materials & methods
No section on statistical analysis. This is a critical omission. L214: should be Green & Craig (1999)

L240: should be Green et al. (2022)

L246-249: Remove if no DNA analysis was conducted

L253: Size class for maturity should be differentiated according to giant clam species, as different species reach maturity differently (Refer to Mingoa-Licuanan and Gomez, 2007, Neo et al., 2019, and Lee et al., 2024). If this does not differentiate species, the size class bin is also inaccurately placed. Recruit is < 5 cm, immature is 5 ≤ 11 cm, do you mean 5 to 11 cm? And mature > 12 cm, then this would omit sizes between 11 cm to 12 cm?

L265: What is the estimated depth for the snorkel survey?

Results
L282: What does this “survey restrictions of high abundance” mean?

L291: Since no statistical test has been mentioned, what do these values mean?

L295: Please be consistent with the unit of measurement. Either claim per hectare or individual ha-1, not both.

L332: What does this statement refer to? Please indicate which figure.

L380: This is rather key information to reveal and warrant a figure with pictures of all three species. The authors can refer to Neo et al 2019, Lee et al. 2022 and Rehm et al. 2022 for providing pictures of the giant clam. I noticed there are JPG files with a description of Title_Image, but these are not embedded in the review PDF file. Those are nice shots of the giant clam.
Please consider compiling those pictures (include all three species) and preparing them into a figure.

Figure and Caption

Figure 2: The Y-axis label is clams/ha, and the caption mentions the number of individual ha-1. Please check throughout the manuscript and be consistent in the unit of measurement used. Use either one, not both; it gets confusing for the reader.

Figure 3: The label shows mean/ha and the caption shows mean density (number of individuals ha-1), but the previous section is clams per hectare. Please be consistent.

Figure 4: mean/ha, is it the same as individual per hectare or individual ha-1? Without the consistency in the unit used, it is very hard to assess the result.
For Figure 4b, try to make the first barplot look bigger by cutting the y-axis or simply put a label to indicate which protection level. And I assume the color polygon is referring to the abundance classes? This needs to be indicated clearly in the caption to lead the reader. Also, make sure the unit is correctly superscript, not ha-1 but ha-1

For Figure 4c, it is very hard to figure out if the n=1 is referring to the number of individuals measured, or number of sites, or even the number of surveys? It would be much easier to interpret the result if it is presented in a histogram or ridge plot (can refer to Lee et al. 2024).

Figure 5. No consistency between figure and caption (number ha-1 vs mean/ha). The plot is very hard to assess, and inaccurate presentation of data. First, the line graph does not make sense, since, x-axis is a factor (location) rather than temporal. Second, size distribution would be better presented in a histogram or a ridgeline plot with histogram, as it would indicate frequency at each size class or bin.

Table and caption

Table 3: The order of citation should be in chronological order (Braley 1987, Green et al. 2022). Please check citation format
The size classes mentioned here are different from the text; please check and verify which one is correct. (In the text is Recruit < 5cm, Immature 5 ≤ 11 cm, and Mature > 12cm). What is count referring to? If the count of individuals, then why is the density 100 for recruits in Aunuu?

Discussion
L442-443: What does bringing down mean?

L445-450: Assessing size classes over time with the line graph is misleading, since it is compared using mean clam size. In fact, there is no reason to compare giant clam size temporally unless it is presented demographically (with size class distribution in a histogram). The author also admitted that those mean values are based on living and dead clams. This section should be voided unless the author clearly separates living and dead clams.

L457: It is not clear how the author stated remoteness and village enforcement stand out as a viable strategy. Does village enforcement mean subsistence? If looking at Figure 4b, village-protected protected have the lowest compared to remote (in red) and subsistence & remote (in green). And the problem is with Figure 4b what does the number indicate in the bar? Is it the number of sites? If it is a number of sites, then the variation in density is likely affected by low sampling effort (size). And with this caveat, the authors cannot recommend any viable strategy at all.

L469-481: This is irrelevant to the study, please consider removing it. If there is a further scuba-based survey still ongoing, the Muliava dataset should be removed. The current dataset on Muliava is based on a snorkel-based survey with a rough estimation of abundances and the area, which might lead to inaccurate estimation and difficult to make direct comparisons with other locations.

L491-505: T. noae and T. maxima are both boring species. And the authors mentioned the opportunistic free-swimming abundance of T. noae in shallow habitats less than 10 feet. This already indicates a potential different ecological niche among the three giant clam species, with T. noae the shallow species, T. maxima the intermediate, and T. squamosa the deep species. But I would remind the authors to rephrase the sentences or refrain from making such a statement, since it is only based on opportunistic surveys with no known survey areas and no species-specific data presented in the manuscript.

L532: No unit mentioned

L536-547: While it is ok to refer to a study on village-based marine protection in general, but current citation is not relevant to giant clams at all. There are several studies with mention village-based or community-based protection on giant clam populations, such as Requilme et al. 2021, Ramah et al. 2019 and etc.

L558-570: While it is good to provide future study directions, it can be further concise into a few sentences.

Conclusions
The conclusion is too long to be a conclusion. Conclusions should be straightforward and concise. The authors mentioned multiple key recommendations, which is good, but if it is too long, it will become a wishlist rather than an action plan for future directions. There is also redundancy and repeated statements made here, which should be incorporated into the Discussion section instead, such as on the expansion of the survey effort.

Supplementary file
The caption for the supplementary table and file is missing. Please provide a detailed description for each table, especially for Supplementary Table S4.

Reference
Please check carefully all the formatting, especially the species name and the article title. All references should be checked manually, even with the use of reference management software.

·

Basic reporting

-

Experimental design

1. Line 251-253: The authors mentioned that they classified the clams as Recruit, Immature and Mature based on shell length and reproductive status while the only data collected for all clams during these surveys were the size (shell length; line 246, 273), photo documentation (lines 246, 272), notes on morphology (e.g., siphon, papillae; lines 246-247, 273-274), small mantle tissue biopsy for DNA analysis (line 248 ) and there is no mention of any examination for gonads (through dissection or biopsy) or using any other direct method (e.g., histology, spawning observation) to assess the clam reproductive status? Moreover, while the size (shell length) is often correlated with maturity, it is not the most appropriate for this situation as factors such as nutrition, temperature, disease, or local environmental conditions can cause variation in the reproductive cycle and size at maturity, so maybe a large clam could be immature; a small clam could be mature under stress and the environmental changes. Could you explain?

2. Line 230: The authors mentioned that they surveyed six islands, but here they listed only five islands (Tutuila, Ofu, Olosega, Ta'ū, Muli va; lines 215-216). Please write the name of the sixth island.

3. Lines 226-228 and 256-257: The authors mentioned the same information about Tutuila surveys/weather. I think this is repeated information.

Validity of the findings

-

Reviewer 3 ·

Basic reporting

The paper “Status and trends of giant clam populations demonstrate the effectiveness of village-based protection in American Sāmoa” is timely and relevant considering the continuous population decline in most of their distribution range, as also reported in Palawan, Philippines (see Dolorosa et al. 2024) and in Malaysia (Lee et al. 2024).

While the government implemented measures to counter this decline, enforcement effectiveness and community cooperation remain a challenge; thus, the a need to understand the dynamics revolving around giant clam conservation that this paper is trying to address.

The paper is well written, needing minor edits such as unclear statements, scientific names that should be written in italics form statements needing citations, and some paragraphs in the Discussion that are supposed to be placed under the Results Section.

Line 41, 42, 131, 133, etc. Please italicize the scientific names
Line 73. The statement “widespread population decline” may cite the works of Lee et al. 2024 and Dolorosa et al. 2024)
Line 215-217. The statement “We surveyed additional sites to Green & Craig around larger islands to gather a more geographically comprehensive distribution” seems unclear to me. This could be restated as “Using the methods of Green & Craig (1999), we surveyed additional sites around larger islands to gather…”
Lines 316-317: Only the data for Open Access is provided. The statement could be more engaging if the authors also present the density data from Protected Sites.

Line 337. Cited Figure 4b, while Figure 4a is cited at a much later part of the paper at line 525. Consider rearranging the order of arrangements of these images in accordance with their use or citation on the paper
Line 378-379 unclear statement
Line 452 -459 The paragraph is more of a Results than a Discussion
Line 497-502. The author argued that T squamosa prefers to inhabit deeper habitats, the reason for its absence in shallow areas. However, T squamosa has been observed to occur abundantly in shallow waters in the Philippines in the past, prior to the massive harvesting. This suggests that their absence in shallow areas is due to exploitation rather than habitat preference.
Lines 525-534 The paragraph is more of a Results than a Discussion
Overall, the paper could be accepted with minor revisions.

References that the authors may add to their paper.

Dolorosa, R. G., Mecha, N. J. M. F., Bano, J. D., Ecube, K. M. A., Villanueva, E. G., & Cabaitan, P. C. (2024). Declining population of giant clams (Cardiidae: Tridacninae) in Palawan, Philippines. Ocean and Coastal Research, 72, e24010.
Lee, L. K., Neo, M. L., Hii, K. S., Gu, H., Chen, C. A., Lim, P. T., & Leaw, C. P. (2024). Vanishing giants: An assessment of the population status of giant clams across Malaysia. Regional Studies in Marine Science, 74, 103546.

Experimental design

.

Validity of the findings

.

Additional comments

.

---

## Round 0.2 · Minor Revisions

· Academic Editor

Minor Revisions

Dear Dr. Marra-Biggs,

Based on the expert reviewers' assessment, some aspects of your valuable article need improvement. You will find the reviewers' comments and suggestions in the attached reports. As you will notice, the reviewers have identified some errors. Therefore, a minor revision is required for your article.

I kindly ask that you revise your manuscript according to the reviewers' suggestions.

Sincerely,

Reviewer 1 ·

Basic reporting

The authors have made considerable revisions to the original manuscript, which have notably improved its readability and clarity. I particularly enjoyed the section discussing the cultural significance of the giant clam (faisua) in American Samoa. The previously omitted statistical analysis section has also been added.

At its current stage, the manuscript is worthy of acceptance, provided that the minor comments listed below are promptly addressed. I therefore recommend acceptance.

Experimental design

With the added statistical analysis section, the authors have satisfied the criteria.

Validity of the findings

With the added statistical analysis section, the authors have satisfied the criteria.

Additional comments

Some minor comments:
L69-70: Please cite the original articles on this statement, Waters et al. (2013) work is on recruitment methodology, not on algal bloom. Also, this statement probably needs rephrase or tone down. Giant clam may be effective in nutrient recycling but there is no direct evidence that presence of giant clams actually prevents eutrophic algal blooms.
L85: Consider citing Lee et al. 2024
Lee, L. K., Neo, M. L., Hii, K. S., Gu, H., Chen, C. A., Lim, P. T., & Leaw, C. P. (2024). Vanishing giants: An assessment on the population status of giant clams across Malaysia. Regional Studies in Marine Science, 74, 103546.

L302-306: Please consider citing the R package that been employed in statistical analysis. R package also gone through careful development much like research articles, hence credit should be given where credit is due.

·

Basic reporting

I have reviewed the revised version of the manuscript and I am satisfied that the authors have addressed all the comments. I have no further comments, and I think the manuscript is ready for publication.

Experimental design

No comments

Validity of the findings

No comments

Additional comments

No comments

Reviewer 3 ·

Basic reporting

My only comment on the revised version of the paper is the use of "feet" as a unit of measure. The author inconsistently used "ft" or "feet" in the manuscript. Because the use of the English System is no longer used in scientific papers, I suggest changing the values and units into "meters or m" (Metric System).

Thank you very much,

Roger

Experimental design

none

Validity of the findings

no more comments

Additional comments

none

---

## Round 0.3 · accepted · Accept

· Academic Editor

Accept

Dear Dr. Marra-Biggs

I thank you for making the corrections and changes requested by the reviewers. I read and checked your valuable article carefully and am happy to inform you that the article has been accepted for publication in PeerJ.

Sincerely yours,

Reviewer 1 ·

Basic reporting

The authors have fulfilled all the criteria.

Experimental design

The authors have fulfilled all the criteria.

Validity of the findings

The authors have fulfilled all the criteria.

Additional comments

Nil